# Characteristics of layered polar mesosphere summer echoes occurrence ratio observed by EISCAT VHF 224MHz Radar

Shucan Ge[1], Hailong Li[1], Tong Xu[2], Mengyan Zhu[2], Maoyan Wang[1], Lin Meng[1], Safi Ullah[1], Abdur Rauf[l]

[1]School of Electronic Science and Engineering, University of Electronic Science and Technology of China, 610054, Chengdu, China

[2]National Key Laboratory of Electromagnetic Environment, China Research Institute of Radiowave Propagation, 266107, Qingdao, China

*Correspondence to*: Hailong Li (hailong703@163.com)

**Abstract.** Polar Mesosphere Summer Echoes (PMSE) are strong radar echoes observed in polar mesopause during the local summer. Observations of layered PMSE carried out by the European Incoherent Scatter Scientific Association Very high frequency (EISCAT VHF) radar during 2004-2015 in the latest solar cycle is used to study the variations of PMSE occurrence ratio (OR). Different seasonal behavior of PMSE is found by analyzing the seasonal variation of PMSE mono-, double- and tri-layer OR. A method was used to calculate the PMSE mono-, double- and tri-layer OR under different electron density threshold. In addition, a method to analyze the correlation of layered PMSE OR with solar 10.7 cm flux index ($F_{10.7}$) and geomagnetic K index is proposed. Based on it, the correlation of layered PMSE OR with solar and geomagnetic activities is not expected to be affected by discontinuous PMSE. It is found that PMSE mono-, double- and tri-layer OR are positively correlated with the K index. The correlation of PMSE mono- and double-layer OR with $F_{10.7}$ is weak, whereas the PMSE tri-layer OR shows a negative correlation with $F_{10.7}$.

**Keywords:** Polar Mesosphere Summer Echoes; EISCAT VHF radar; solar 10.7 cm flux index ($F_{10.7}$); geomagnetic K index

## 1 Introduction

The ionosphere is an important part of near the Earth space environment, and the mesosphere is the coldest region in the Earth's atmosphere. Polar Mesosphere Summer Echoes (PMSE) are strong echoes

detected by radars from medium frequency (MF) to ultra-high frequency (UHF) bands in polar summer mesopause, and PMSE has been considered to be possible indicators of global climate change (Thomas and Olivero, 2001). The observation range is from 75 to 100 km where the strongest echo occurs at the altitude of about 86 km on average (Czechowsky et al., 1979). Radar waves in the very high frequency

(VHF) band are backscattered due to the irregularities of electron density with spatial scales of about half the radar wavelength. It has been confirmed by Blix et al. (2003) from simultaneous rocket and radar observations. The most extensively accepted theory is that the irregularities of electron density are sustained due to the reduction in electron diffusion characterized by the slowest ambipolar diffusion mode associated with the charged ice grains (Cho et al., 1992). Varney et al. (2011) scrutinized one

particular aspect of the turbulent theory of PMSE: the electron density dependence of the echo strength. One remarkable feature of all PMSE is the fact that the radar echoes often occur in the form of two or more distinct layers which can persist for periods up to several hours. Until now, the layering mechanism leading to these multiple structures is only poorly understood in spite of some previous attempts involving gravity waves, the general thermal structure, and Kelvin-Helmholtz-instabilities (Röttger, 1994;

Klostermeyer, 1997; Hill et al., 1999, Hoffmann et al., 2005).

Palmer et al. (1996) statistically analyzed the PMSE in northern hemisphere observed by the EISCAT VHF radar during 1988-1993. They suggested that: (1) PMSE are summer phenomena, lasting from June to August; (2) PMSE occur mostly around noon and midnight, following a semidiurnal pattern; (3) the echoing structures move bodily, perhaps in response to gravity waves. Based on measurements at

20 Andenes, Norway, observed by the 53.5 MHz ALOMAR SOUSY radar during 1994-1997 and the ALWIN radar during 1999-2001, Bremer et al. (2003) found that the variation of PMSE is markedly controlled by solar cycle variations and precipitating high energetic particle fluxes. Bremer et al. (2006) discussed that the strength of PMSE depends on the level of ionization because of the long-term changes of mesospheric summer echoes caused by the incident solar wave radiation and precipitating high

energetic particle fluxes from about 20 May to the end of August during 1998-2006. Smirnova et al. (2010) used the ESRAD MST radar's measurements and found that the inter-annual variations of PMSE OR and length of the season anticorrelated with solar activity ($F_{10.7}$ index, the daily solar activity proxy) but not significant, and PMSE OR correlate with geomagnetic activity (AP index). However, no statistically significant trends in PMSE yearly strengths were found in their work. Smirnova et al. (2011)

concentrated on the accurate calculation of PMSE absolute strength as expressed by radar volume reflectivity and found that the inter-annual variations of PMSE volume reflectivity strongly correlate with the local geomagnetic K index and anticorrelate with solar 10.7 cm flux. However, they did not find any statistically significant trend in PMSE volume reflectivity during 1997-2009. Li and Rapp (2011) reported that PMSE OR at 224 MHz shows a positive correlation with both the solar and geomagnetic activities. PMSE have been detected and widely studied based on long-term observations of many different MST radars (Reid et al., 1989; Thomas et al., 1992; Smirnova et al., 2011). Since from the first observation of PMSE in 1979, it is well-known that the PMSE observations are different when observed by different frequency radar even at the same sites, and PMSE often shows obvious layered events.

Many studies have widely reported that there is a significant correlation between the ionization level and PMSE observed by 53.5 MHz radar (Inhester et al., 1990; Belova et al., 2007; Latteck et al., 2008). The correlation of the ionization level with PMSE at 224 MHz is as significant as that the correlation of the ionization level with PMSE at 53.5 MHz, then previous studies provide the research basis and ideas for the PMSE study detected by 224MHz radar. There are still a few significant problems that must be solved with the characteristics of layered PMSE OR. Hence, it is necessary to analyze the layered PMSE OR and study layered PMSE characteristics deeply with data measured by 224 MHz EISCAT VHF radar under different observation conditions. The statistical results of layered PMSE OR with the same radar at the same site over the period 2004-2015 are given in this paper, which was based on the experiment data detected by 224 MHz EISCAT VHF radar. In addition, the correlation of PMSE OR with geomagnetic K index and $F_{10.7}$ is analyzed and discussed. The method of the correlation analysis between layered PMSE OR and solar activity and between layered PMSE OR and geomagnetic activity is given in this paper without being affected by the defect of discontinuous PMSE measurements of EISCAT radar. It makes a significant breakthrough in the characterization of the layered PMSE OR. The aim of the current work is to provide definitive data foundation for further analysis and the investigation of the physical mechanism of PMSE.

**2 radar and experiment data description**

The PMSE observations used here were obtained with 224MHz EISCAT VHF radar from 2004 to 2015. EISCAT VHF radar is located at Tromsø, Norway (69.35°N, 19.14°E), using a parabolic cylindrical

120m×40m antenna. It is a powerful tool to study the lower ionosphere. Detailed descriptions of the radar can be found in Baron (1986). The measurements by EISCAT radar are very well suited for investigating the characteristics of PMSE (for previous work, see e.g. Li et al., 2010 and references therein). It has frequency and phase modulation capability with pulse length of $1\,\mu s$ to $2\,ms$. The parameters are shown in Table 1 for accuracy control of EISCAT VHF radar.

EISCAT VHF radar ran several standard experiment modes: "manda, beata, bella, tau7, arcd (arc_dlayer) and tau1". The main differences between these experiment modes are illustrated in Table 2. The manda and arcd modes mainly used for low altitude detection and provide spectral measurements at mesospheric altitude. Therefore, the accurate data used in this study is mainly provided by manda and arcd modes.

. **Table 1** Parameters of the radars.

| Radar | EISCAT VHF |
|---|---|
| Location | 69.59º N 19.23º E |
| Operating frequency | 224 MHz |
| Transmitter peak power | 1.5 MW |
| Antenna 3-dB beam width | 1.7º NS × 1.2º EW |
| Antenna effective area | 5690 m$^2$ |
| Pulse length (altitude resolution) | 300 m |
| Pulse repetition frequency | 741 Hz |
| No. of bits in code | 64 |
| No. of code permutations | 128 |
| No. of coherent integrations | 1 |
| Lag resolution | 1.35 ms |
| Maximum lag | 0.17 s |

**Table 2** EISCAT VHF radar standard experiments.

| Name | Code length [bit] | Baud length [μs] | Sampling rate[μs] | Range span[km] | Time resolution [s] | Plasma line | Raw data |
|---|---|---|---|---|---|---|---|
| manda | 61 | 2.4 | 1.2 | 19–209 | 4.8 | - | Yes |
| arc_dlayer | 64 | 2 | 2 | 60–139 | 5.0 | - | - |
| beata | 32 | 20 | 20 | 52–663 | 5.0 | Yes | - |
| bella | 30 | 45 | 45 | 63–1344 | 3.6 | Yes | - |

| tau7 | 16 | 96 | 12 | 50–2001 | 5.0 | - | - |
| tau1 | 16 | 72 | 24 | 104–2061 | 5.0 | - | - |

## 3 Data analysis

In this study, we use the EISCAT VHF radar data from 2004 to 2015. The software package GUISDAP (Grand Unified Incoherent Scatter Design and Analysis Program) (see Lehtinen and Huuskonen, 1996 and www.eiscat.se for details) was used to analyze the radar data. The electron density $N_e$ analyzed by GUISDAP software was obtained between $10^6$ and $10^{14}$ m$^{-3}$. The level of electron density represents the intensity of echoes.

First of all, the heating parts were removed from the data set to avoid the heating effect. After that, the presence of PMSE was defined as the threshold of electron density ($N_e > 2.6 \times 10^{11}$ m$^{-3}$). We have used the PMSE threshold given by Hocking and Röttger (1997) and Qiang Li (2011) (see Appendix A Table A.2). Besides, some abnormal echoes are related to meteor. It is not considered to be PMSE and is neglected in later discussion. PMSE is not continuous in time. If the electron density satisfies the threshold ($N_e > 2.6 \times 10^{11}$ m$^{-3}$), we considered it as a PMSE event. We have considered only those events which PMSE echoes are continuous for time (t ≥ 1 min).

## 4 Results

### 4.1 Layered PMSE events

PMSE occur in thin layers having an average thickness up to 3-4 km of the monolayer, and the mean altitude distribution of PMSE events is 80-90km. It is considered to be the area of independent anomalous echoes. Fig. 1 (a), (b) and(c) show the typical events of PMSE monolayer, double-layer and tri-layer, respectively. As mentioned in the introduction, a notable feature of PMSE observed by radar is that the radar echoes typically occur in the form of two or more layers. However, the systematic theories of the layering mechanism led to these multiple structures didn't come into being. Here we will study the

occurrence of these layered PMSE events and their relationships with solar and geomagnetic activity. This content will be discussed in detail later in the paper.

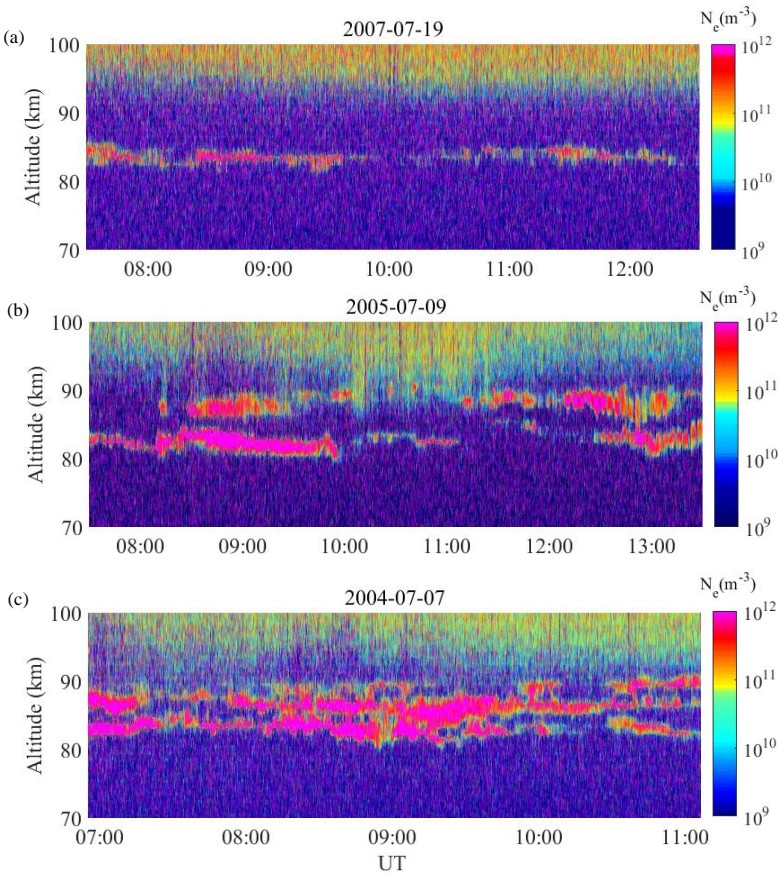

**Fig. 1 The typical layered PMSE events observed by EISCAT 224MHz VHF radar. a) Monolayer PMSE; b) Double layer PMSE; c) Tri-layer PMSE.**

## 4.2 Layered PMSE OR calculation method

The calculation method is based on individual horizontal profiles. When the electron density satisfies the

PMSE threshold ($N_e > 2.6 \times 10^{11} m^{-3}$), then that time was taken as the starting time of the PMSE occurrence and until the time when the electron density fails to satisfy the threshold was taken as the end time of PMSE occurrence. The time of PMSE duration is the time difference between the end and the starting time of the PMSE occurrence. The time interval not be regarded as PMSE occurrence time, if the time interval between them is shorter than 1 minute (t < 1 min). Taking the calculation method of monolayer

PMSE OR as an example: We defined the ratio between the sustained time of monolayer PMSE and the

total observation time as the monolayer PMSE OR. The applied procedure for the detection of multiple

PMSE layers is based on individual vertical profiles with a high temporal resolution (Hoffmann, 2005).

The layer ranges are identified by an electron density threshold of $2.6 \times 10^{11} m^{-3}$ ($N_e > 2.6 \times 10^{11} m^{-3}$). Once

a vertical profile of the electron density has two peaks and these two peaks are higher than the threshold

5    ($N_e > 2.6 \times 10^{11} m^{-3}$), we select it as a double layer. The PMSE double-layer OR is the ratio between the

sustained time of PMSE double layer and the total observation time. The tri-layer OR is also calculated

by the same way.

**4.3 The variations of layered PMSE occurrence ratios**

The layered PMSE OR, layered PMSE occurrence time (OT) and total observing time detected by

10    EISCAT VHF radar from 2004 to 2015 are illustrated in Table 3. PMSE mono-, double-, tri-layer and

total OR are also presented in Table 3.

**Table 3** Statistical data from 2004 to 2015.

| Year | Total Observing Time (min) | Monolayer PMSE OT (min) | Double Layer PMSE OT (min) | Tri-layer PMSE OT (min) | Monolayer OR [%] | Double layer OR [%] | Tri-layer OR [%] | Total OR [%] |
|---|---|---|---|---|---|---|---|---|
| 2004 | 16054 | 4701 | 2774 | 151 | 29.28 | 17.28 | 0.94 | 47.50 |
| 2005 | 8165 | 3564 | 1491 | 182 | 43.65 | 18.26 | 2.23 | 64.14 |
| 2006 | 9248 | 2950 | 910 | 93 | 31.78 | 9.84 | 1.01 | 42.63 |
| 2007 | 9341 | 3027 | 804 | 0 | 32.41 | 8.61 | 0.00 | 41.02 |
| 2008 | 3310 | 763 | 97 | 0 | 23.06 | 2.92 | 0.00 | 25.98 |
| 2009 | 2264 | 424 | 76 | 8 | 18.72 | 3.34 | 0.35 | 22.41 |
| 2010 | 6303 | 1799 | 498 | 53 | 28.54 | 7.90 | 0.84 | 37.28 |
| 2011 | 9638 | 3624 | 2692 | 202 | 37.60 | 27.93 | 2.10 | 67.63 |
| 2012 | 7497 | 3550 | 1554 | 207 | 47.35 | 20.73 | 2.76 | 70.84 |
| 2013 | 14037 | 6906 | 3873 | 532 | 49.20 | 27.59 | 3.79 | 80.59 |
| 2014 | 2971 | 998 | 731 | 64 | 33.60 | 24.6 | 2.15 | 60.35 |
| 2015 | 4776 | 2019 | 1022 | 22 | 42.28 | 21.40 | 0.46 | 64.14 |

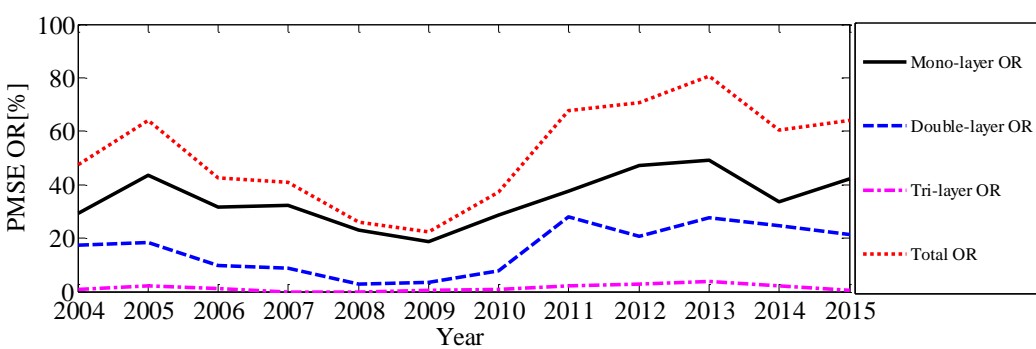

**Fig. 2 Annual mean layered PMSE occurrence ratio. The OR of total (red dot line). The OR of monolayer (black solid line). The OR of double-layer (blue dashed line). The OR of tri-layer (pink dot-dashed line).**

Fig. 2 shows that the annual mean mono- double- and tri-layer OR agrees with the total PMSE OR. We calculated the correlation of the annual mean mono-layer with double-layer OR, tri-layer OR and total OR using the Spearman rank correlation coefficients (It will be particularly described in section 4.3.2). The correlation coefficients ($r_s$) of mono-layer with double-layer OR, tri-layer OR and total OR are 0.7922, 0.7718 and 0.9480, respectively. All the correlation coefficients are statistically significant with $P<0.05$. These high values of correlation coefficients show that the correlation of annual mean mono-layer with annual mean double-layer OR, tri-layer OR, and total OR is very high. In addition, the annual mean layered PMSE OR from 2008 to 2010 is relatively low, and the solar activity is relative 'quiet' in these years.

Fig. 2 shows two significant phenomena: (1) The variation trends of annual mean mono-, double- and tri-layer PMSE OR has rules to follow, i.e., the OR of monolayer is the highest, double-layer lies in the middle and the tri-layer is the lowest. (2) The annual mean layered PMSE and total OR values show a similar shape of the sinusoidal, which has obvious wave peak and wave valley. One wave peak lies in 2005, and the other lies in 2013. The values of two wave peaks are different and the values in 2005 are smaller than that in 2013. The values of the wave valley lie in 2008-2009. Here we only give the results of the data analysis, no longer do the cause analysis, because the stratification of PMSE is affected by many factors and hasn't been decided yet. The analysing method and results given in this paper have a significant reference value for studying the PMSE phenomenon.

## 4.4 Seasonal behavior

The mean seasonal variations of the layered PMSE OR and PMSE total OR observed by EISCAT VHF radar during 2004-2015 is shown in Fig. 3 and Fig. 4, respectively. Fig. 3 illustrates the mean seasonal variation of the mono- (blue bars) double- (yellow bars), tri-layer (red bars) PMSE OR and quartic polynomial fitting for the monolayer PMSE OR (black dot-curve) during 2004-2015. Fig. 4 shows the mean seasonal variation of PMSE total OR (blue bars) and $3/\pi$ harmonic fitting for total PMSE OR (black dot-curve) during 2004-2015. It is clear from Fig. 3 and Fig. 4 that the monolayer PMSE in the Tromsø, Norway, often begins in late May, reaches its maximum in early June or mid-June, keeps this level until the end of July or beginning of August, and gradually decreases or vanishes when it is close to the end of August or the beginning of September in general which is in agreement with Smirnova et al., (2011). The double-layer PMSE also begins in late May, but its maximum value appears in mid-July. In addition, it keeps the larger value in June and July, and it simply fades away in early August. The tri-layer PMSE appears a lot less in comparison with mono- and double- layer PMSE. In terms of time, it appears later and disappears earlier. Furthermore, the tri-layer PMSE OR is large at the end of June and early July, which is different from monolayer and double layer PMSE OR.

According to the statistical results, monolayer, double-layer and tri-layer PMSE OR have seasonal variation. Moreover, there is fluctuation in the trends of $F_{10.7}$ and geomagnetic K index. Therefore, it is necessary to investigate the correlation of solar and geomagnetic activity with different layered PMSE OR during 2004-2015, and we should try to explain the occurrence mechanism of PMSE. It is well known that other missions apart from PMSE regular observations are performed by EISCAT VHF radar, so EISCAT radar does not provide continuous PMSE observations. We raise an important question: Table 3 indicates a difference in total observation time for the individual years. How has this been taken into account for the determination of occurrence ratios? To solve this problem, we use another method to recalculate the layered PMSE OR. Then, the correlation between the layered PMSE OR and the $F_{10.7}$ and between the layered PMSE OR and K index are studied. As mentioned in the calculation method section, we only select the days when PMSE presents and calculate the layered OR of PMSE.

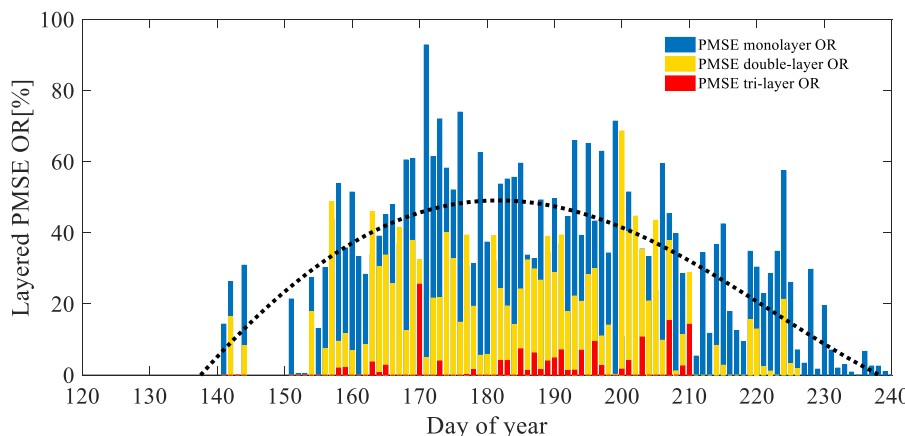

**Fig. 3 Mean seasonal variation of mono-(in blue), double-(in yellow), tri-layer (in red) PMSE occurrence ratio from 2004 to 2015.**

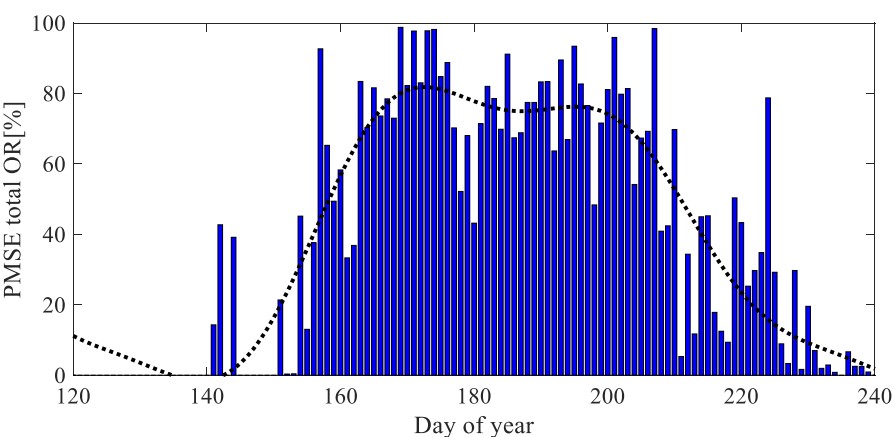

**Fig. 4 Mean seasonal variation of total PMSE occurrence ratio.**

## 5 Discussion

The layered PMSE OR was calculated and the relations among PMSE mono-, double- and tri-layer OR were analyzed statistically. At the same time, the mean seasonal variations of the layered PMSE OR and PMSE total OR have been presented. Hoffmann (2005) shows that the layering occurs because of subsequent nucleation cycles of ice particles in the uppermost (and coldest) gravity wave induced temperature minimum (see Hoffmann, 2005, Figure 3a). Subsequently, these newly created ice particles grow and sediment down and lead to the distinct layering. Besides, Rapp and Lübken (2004) found that charged ice particles and atmospheric turbulence play major roles in the change of the electron number density that leads to PMSE in the mesopause region. We know that solar and geomagnetic activities have

a certain degree of influence on the occurrence of PMSE, however, the effects of solar and geomagnetic activities on layered PMSE are not understood well. Therefore, it is necessary to study the effects of solar and geomagnetic activities on layered PMSE. The occurrence ratio obtained by the ratio of the occurrence time of PMSE to the total observation time is the calculation method in the traditional sense. It is easy to understand and accurately analyze the short-term variations, such as diurnal variation and seasonal variation of PMSE. However, the long-term trend is subject to error and dispute by this calculation method. Furthermore, it is difficult to discuss and analyze the correlation of layered PMSE OR with solar and geomagnetic activities. Therefore, we have presented a new calculation method for calculating the layered PMSE occurrence ratio, which is different from the method given in section 4.2. So that, the layered PMSE OR is relatively accurate. The correlation of PMSE with solar and geomagnetic activities is not expected to be affected by discontinuous PMSE. The study of relations between PMSE and solar activities and between PMSE and geomagnetic activities are significative.

## 5.1 Another method for layered PMSE OR Calculation

The emphasis of this section is to present a hybrid algorithm based on grid partitioning. The calculation method is based on altitude. A large number of literatures and experimental observations have shown that the altitude range of PMSE is 80-90km (Li and Rapp, 2011; Smirnova et al., 2010; Latteck and Bremer, 2013). Hoffmann (2005) shows a mean height of 84.8 km for monolayer PMSE. In the case of multiple layers PMSE, the lower layer occurs at a mean height of ~83.4 km. The second layer in the case of multiple PMSE layer structures shows a maximum at about 86.3 km (The judging criteria in regard to the multiple layer PMSE see section 4.3). Firstly, we counted the total number of electron density at altitude of 80-90km and then counted the number of electron density satisfying the PMSE threshold ($N_e$ $>2.6\times10^{11}\text{m}^{-3}$) in the period when the PMSE is known to be present (if electron density satisfies the threshold $N_e>2.6\times10^{11}\text{m}^{-3}$, we identify layered PMSE exist at this moment). The ratio between the numbers of layered PMSE electron densities values larger than the threshold and the numbers of total electron density at altitude of 80-90 km was calculated. The double-layer and tri-layer PMSE OR calculated by this method is higher than the layered PMSE OR calculated by the method given in section 4.2. The correlation coefficients were calculated between PMSE OR and the 10.7cm of the solar flux index ($F_{10.7}$) and between PMSE OR and geomagnetic K index, respectively. The PMSE have been

identified only for the time of PMSE duration lager than 1 min (t≥1 min). Because the integration time of manda and arcd models are 4.8s and 2s respectively, on the basis of the condition (t≥1 min), the PMSE is needed to be for≥12 and 30 data points, respectively.

### 5.2 Layered PMSE OR under different electron density threshold

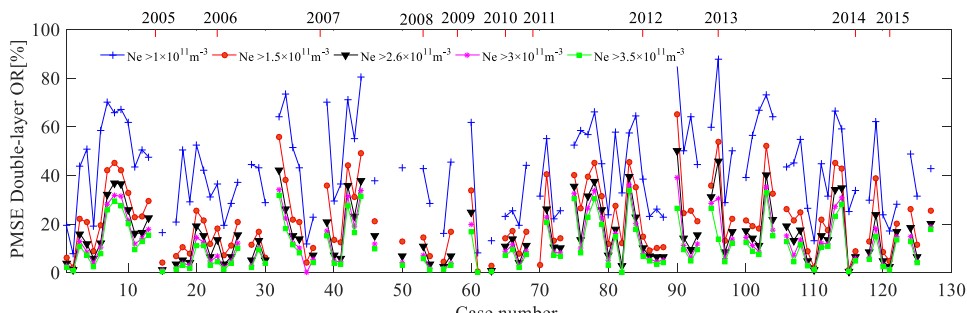

**Fig. 5 PMSE monolayer occurrence ratio under different electron density threshold with axis at the top showing the time in years.**

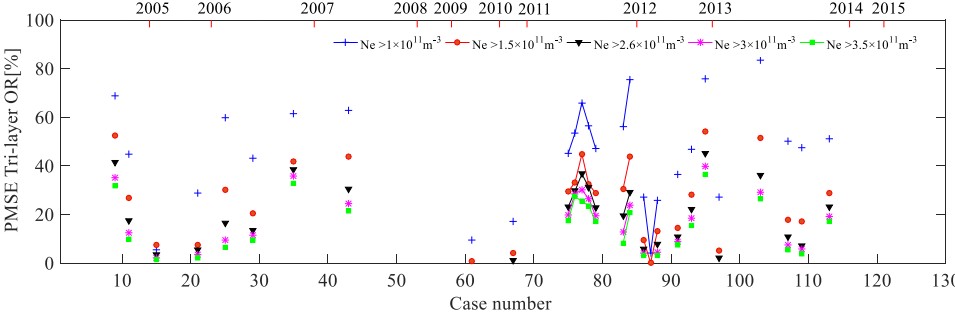

**Fig. 6 PMSE double-layer occurrence ratio under different electron density threshold with axis at the top showing the time in years.**

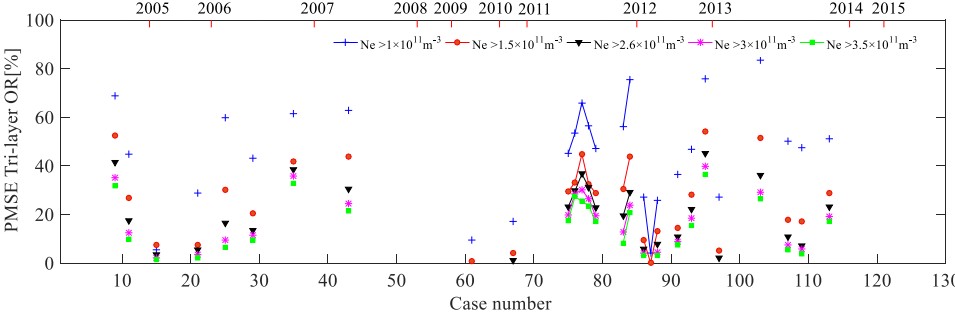

15 **Fig.7 PMSE tri-layer occurrence ratio under different electron density threshold with axis at the top showing the time in years.**

In this section, the day when the first occurrence of PMSE in 2004 (regardless of duration) was recorded as 1, and the day with the later occurrence of PMSE increased by sequence. Using this sequence as the horizontal axis and layered PMSE OR with different electron density threshold as the vertical axis, the results are shown in Fig. 5, 6, and 7. That is, Fig. 5, Fig. 6 and Fig. 7 show PMSE mono- double- and tri-layer OR under different electron density threshold, respectively. In the calculation method section we have defined the electron density threshold ($N_e > 2.6 \times 10^{11} \text{m}^{-3}$). Here, we give the layered PMSE OR with threshold $N_e > 1 \times 10^{11} \text{m}^{-3}$, $N_e > 1.5 \times 10^{11} \text{m}^{-3}$, $N_e > 2.6 \times 10^{11} \text{m}^{-3}$, $N_e > 3 \times 10^{11} \text{m}^{-3}$ and $N_e > 3.5 \times 10^{11} \text{m}^{-3}$, respectively. We found that the variation trends of layered PMSE OR with different threshold are largely consistent. In addition, the larger the threshold, the smaller the ratio. Smirnova et al. (2010) analyzed day-to-day and year-to-year variations of PMSE OR for different thresholds. They found that the choice of the threshold does not influence the shape of the variation curves for PMSE OR. Zeller and Bremer (2009) indicated that different threshold values are for the investigations of the influence of geomagnetic activity on PMSE, however, of less importance. They both think that the variation trends of PMSE OR with different threshold are consistent. The aim of choosing 5 different thresholds is also to increase the number of samples for calculating the correlation coefficients between layered PMSE OR and $F_{10.7}$ and between layered PMSE OR and K index. Since these occurrence ratios are calculated in the case where the occurrence of PMSE is determined, it is recognized that these occurrence rates are reliable. It is well known that the period of 2006-2009 is solar minimum and 2012 is solar maximum, but the PMSE mono- and double-layer average OR in 2007 is not consistent with solar activity. In other words, there is no obvious correlation between mono- and double-layer PMSE OR and solar activity. Besides, we found that tri-layer PMSE OR and solar activity are in opposite directions. To prove the conclusion, we will calculate the correlation coefficient between layered PMSE OR and solar activity and between layered PMSE OR and geomagnetic activity in the next section. Therefore, the correlation between them can be judged directly.

### 5.3 Effect of solar and geomagnetic activity on PMSE OR

### 5.3.1 $F_{10.7}$ index and K index

The $F_{10.7}$ index is a measure of the solar radio flux per unit frequency at a wavelength of 10.7 cm, near the peak of the observed solar radio emission. $F_{10.7}$ is often expressed in SFU or solar flux units (1 SFU = $10^{-22}$ W·m$^{-2}$·Hz$^{-1}$). It represents a measure of diffuse, nonradiative coronal plasma heating. It is an excellent indicator of overall solar activity levels and correlates well with solar UV emissions. The K-index quantifies disturbances in the horizontal component of Earth's magnetic field with an integer in the range 0-9 with 1 being calm and 5 or more indicating a geomagnetic storm. It is derived from the maximum fluctuations of horizontal components observed on a magnetometer during a three-hour interval. The K-index was introduced by Julius Bartels in 1939(Bartels et al., 1939). The K index values used in the paper is the median of the K index observed on a magnetometer during a day, where the effect of the heating experiments was removed.

### 5.3.2 Correlation coefficients

A correlation coefficient is a numerical measure of some type of correlation, meaning a statistical relationship between two variables (Boddy and Smith, 2009). The Pearson correlation coefficient known as Pearson's $r$, is a measure of the strength and direction of the linear relationship between two variables that is defined as the covariance of the variables divided by the product of their standard deviations. Pearson's correlation coefficient Given a pair of random variables (X, Y), the formula for $r$ is (Wilks, 1995):

$$r_{X,Y} = \frac{\text{cov}(X,Y)}{\sigma_X \sigma_Y}$$

Where:

*Cov* is the covariance.

$\sigma_X$ is the standard deviation of $X$

$\sigma_Y$ is the standard deviation of $Y$.

Spearman's rank correlation coefficient is a measure of how well the relationship between two variables can be described by a monotonic function. The Spearman correlation between two variables is equal to the Pearson correlation between the rank values of those two variables. While Pearson's correlation assesses linear relationships, Spearman's correlation assesses monotonic relationships (whether linear or

not) (Well and Myers, 2003). For a sample of size $n$, the $n$ raw scores $X_i$, $Y_i$ are converted to ranks $rgX_i$, $rgY_i$, and $r_s$ is computed from:

$$r_S = \frac{\text{cov}(rg_X, rg_Y)}{\sigma_{rg_X}\sigma_{rg_y}}$$

Where:

$\quad$ $\text{cov}(rg_X, rg_Y)$ is the covariance of the rank variables.

$\sigma_{rg_X}$ and $\sigma_{rg_Y}$ are the standard deviations of the rank variables.

A high value (approaching +1.00) is a strong direct relationship, values near 0.50 are considered moderate and values below 0.30 are considered to show weak relationship. A low negative value (approaching -1.00) is similarly a strong inverse relationship, and values near 0.00 indicate little, if any

$\quad$ relationship.

To determine whether a result is statistically significant, a $P$-value is calculated which is the probability of observing an effect of the same magnitude or more extreme given that the null hypothesis is true (Devore, 2011). The null hypothesis is rejected if the $P$-value is less than a predetermined level (usually $\alpha=0.05$). Where $\alpha$ is called the significance level, and it is the probability of rejecting the null hypothesis

$\quad$ given that it is true (a type I error).

### 5.3.3 Correlation between layered PMSE OR, $F_{10.7}$ and K index

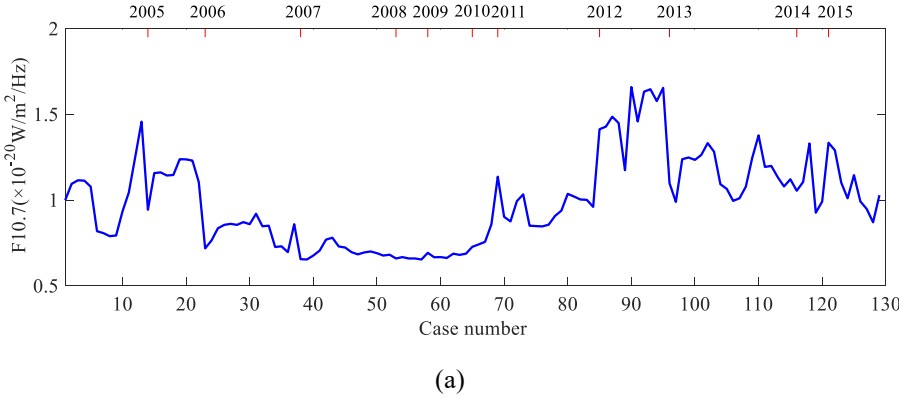

(a)

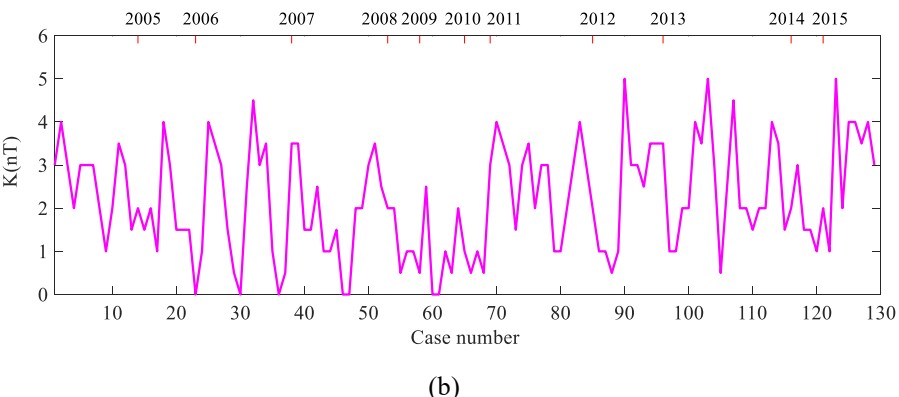

(b)

**Fig. 8 (a) The variations of F$_{10.7}$ values corresponding to the occurrence of PMSE with axis at top showing the time in years. (b) The variations of geomagnetic K index values corresponding to the occurrence of PMSE with axis at the top showing the time in years.**

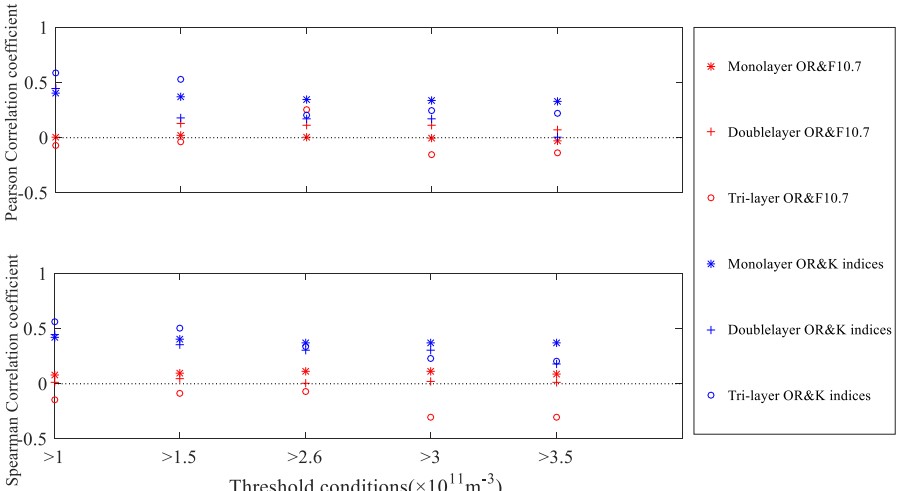

**Fig. 9 Pearson linear and Spearman rank correlation computed between layered PMSE OR (with thresholds $N_e > 1\times10^{11}\text{m}^{-3}$, $N_e > 1.5\times10^{11}\text{m}^{-3}$, $N_e > 2.6\times10^{11}\text{m}^{-3}$, $N_e > 3\times10^{11}\text{m}^{-3}$ and $N_e > 3.5\times10^{11}\text{m}^{-3}$, respectively) and F$_{10.7}$ corresponding to the occurrence of PMSE and between layered PMSE OR and K index corresponding to the occurrence of PMSE, respectively. For each correlation coefficient, $P$ value is less than 0.05. The horizontal dotted line is drawn to separate positive and negative correlation coefficients.**

Fig.8 shows the variations of F$_{10.7}$ and geomagnetic K index values corresponding to the occurrence of PMSE. The correlation of PMSE with solar and geomagnetic activities is not expected to be affected by discontinuous PMSE because of the F$_{10.7}$ and K values corresponding to the occurrence of PMSE with threshold of $N_e > 2.6\times10^{11}\text{m}^{-3}$. So, the study of relations between PMSE and solar activities and between PMSE and geomagnetic activities make sense. The relation between layered PMSE OR and F$_{10.7}$ and between layered PMSE OR and K values can be analyzed for the results shown in conjunction with

Figures 5 through 8. In order to examine the correlation between layered PMSE OR and $F_{10.7}$ and between layered PMSE OR and K index, all the data points of PMSE OR, $F_{10.7}$ and K index with simultaneous occurrence were combined. Fig.9 shows the correlation coefficients computed by combing all the points of PMSE OR (with thresholds $N_e > 1 \times 10^{11} m^{-3}$, $N_e > 1.5 \times 10^{11} m^{-3}$, $N_e > 2.6 \times 10^{11} m^{-3}$, $N_e > 3 \times 10^{11} m^{-3}$ and $N_e > 3.5 \times 10^{11} m^{-3}$), $F_{10.7}$ and K index with simultaneous occurrence, and we apply significant test. It is seen from Fig.9 that layered PMSE OR is positively correlated with the K index and the coefficients indicate a moderate correlation between the variables. Whereas the correlation coefficient between PMSE mono- and $F_{10.7}$, double-layer OR and $F_{10.7}$ both are very low, indicating that their correlation is weak or even irrelevant. Interestingly, we found that the PMSE tri-layer OR has a negative correlation with $F_{10.7}$, although the correlation was lower than what we have supposed. This finding never published in previous literature. Hence, it is indicated that the cases with positive values play a decisive role when calculating the correlation coefficient between the data points of PMSE and K index occur simultaneously, and events with negative values dominate in the calculation of the correlation coefficient between tri-layer PMSE OR and $F_{10.7}$. But mono-, double-layer PMSE OR has rare relevance with $F_{10.7}$.

The correlation between layered PMSE OR and $F_{10.7}$ and between layered PMSE OR and K index have been obtained. It indicates that there are many complicated factors for the formation and development of PMSE besides solar and geomagnetic activities. There are explanations for these results: on one hand, the enhanced solar activity increases the electron density due to the increase of ionization, and with the increase of solar radiation, the photodissociation enhance and the water vapor content is reduced. On the other hand, the positive correlation between PMSE OR and K index may be apprehensible, because the enhanced magnetic activity caused precipitating particles increase in the mesosphere, and lead to increase in electron densities. Latteck and Bremer (2013) show that PMSE are caused by inhomogeneities in the electron density of the radar Bragg scale within the plasma of the cold summer mesopause region in the presence of negatively charged ice particles. Thus, the occurrence of PMSE contains information about mesospheric temperature and water vapor content but also depends on the ionization due to solar electromagnetic radiation and precipitating high energetic particles. However, we still cannot explain why there is a negative correlation between tri-layer PMSE OR and $F_{10.7}$. This should be noticed in future research.

**6 Summary and Conclusions**

In this paper, the PMSE occurrence ratios with monolayer, double- and tri-layers detected by EISCAT VHF radar during a solar cycle have been presented. The daily and seasonal variation of the layered PMSE was analyzed. We implemented a method to provide more accurate conclusions on the study of the long-term variation of PMSE with different thresholds. The correlation between layered PMSE and solar radiation flux ($F_{10.7}$) and between layered PMSE and geomagnetic activity (K index) was given. The following conclusions were reached:

(1)  Mono-, double- and tri-layer PMSE have different seasonal behaviors. Monolayer PMSE often begins in late May, reaches its maximum in early June or mid-June, keeps this level until the end of July or beginning of August, and gradually decreases or vanishes when it is close to the end of August or the beginning of September in general, which is in agreement with the earlier report (Smirnova et al., 2011). The double-layer PMSE OR reaches its maximum in mid-July and simply fade away in early August. The tri-layer PMSE appears later and disappears earlier in comparison with mono-and double-layer PMSE, and it is large in the end of June and early July.

(2)  The variation trends of mono- double- and tri-layer PMSE OR under different electron density thresholds are greatly consistent. It is found that the larger the threshold, the smaller the ratio. Beyond that, PMSE mono- and double-layer OR are not associated with solar activity. PMSE tri-layer OR is inversely proportional to solar activity.

(3)  Layered PMSE OR is positively correlated with the K index. The correlation between PMSE mono- and double-layer OR and $F_{10.7}$ is relatively weak, and PMSE tri-layer OR has a negative correlation with $F_{10.7}$.

*Data availability.*
All    EISCAT    data    used    in    this    work    have    been    downloaded    at
https://www.eiscat.se/schedule/schedule.cgi.
*Competing interests.* The authors declare that they have no conflict of interest.

**Authors' contributions**

Shucan Ge designed this study, carried out statistics, analyzed the results and wrote the manuscript. Hailong Li participated in the design of the study and the analysis of the results. Tong Xu and Mengyan

Zhu helped with the conceptual ideas for the paper. Maoyan Wang and Lin Meng managed this study and participated in language grammar modification. Safi Ullah and Abdur Rauf participated in modifying language issues and provided a lot of suggestions about the revised manuscript. All authors read and approved the final manuscript.

**5    Acknowledgments**

This study is supported by the National Natural Science Foundation of China [No. 41104097 and No.41304119]. This study is also supported by the National Key Laboratory of Electromagnetic Environment, China Research Institute of Radiowave Propagation (CRIRP). We also acknowledge EISCAT, which is an international association supported by China, Finland, Japan, Norway, Sweden, and the UK. I would like to thank Wen Yi who gave us valuable opinions and suggestions for the revised manuscript.

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
