# Peer review of "Characteristics of layered polar mesosphere summer echoes occurrence ratio observed by EISCAT VHF 224MHz Radar"

_Annales Geophysicae, 2019_

## Referee Comment (RC1) · Anonymous Referee #1 · 4 Mar 2019

General comments

The paper by Shucan Ge et al. investigates the occurrence Polar Mesosphere Summer Echoes (PMSE) over a solar cycle. Besides the statistical study of PMSE occurrence, the authors propose a method to deal with the discontinuous EISCAT radar measurements. They argue that such method makes easier to establish a relationship between the PMSE and the solar and geomagnetic activities. The paper presents a relatively large data set that brings results which are worth to be published. However, there are still points to be addressed. I recommend minor revision prior the publication.

Specific comments

[Figure]

The manuscript needs some editing of English. Sometimes it is difficult to follow and understand it. Several times the authors employ commas instead of using an end point to finish the idea expressed in the sentence. I recommend revising the writing style.

At page #1, lines 1-2: the sentence: "The ionosphere is an important part of the near the earth space environment and the mesosphere is the coldest region in the earth's atmosphere at local summer time." Regardless the season the mesosphere is the coldest region of the Earth's atmosphere, not only during the summer. I suggest rewrite the sentence to make this clearer. Section 3.1-Calculation method

The authors should explain better the reason to use the threshold of 2.6x1011 electrons/m3 to detect the PMSE.

At page #6, lines 11-13: In that sentence the authors mention a condition t $\geq$ 1 min. It is not clear where this condition came from. They should make this clearer.

The description of the method of calculation at page #6, lines 15-20, which takes as an example of the monolayer PMSE occurrence, seems to be a little confused. The description is clearer when the authors described the occurrence ratio of the double and tri-layer PMSE. I suggest to rewrite the description of the monolayer PMSE occurrence.

From Table 3 one can see that the author defined the OR of the PMSE as the percentage ratio between the duration of the mono, double and triple layer PMSE and the total time of observations. The description mentioned above should be as clear as the information coming from the Table 3.

In section 4, the authors propose a method to make PMSE OR continuous. They considered as day 1 the first PMSE occurrence in 2004, day 2 as second PMSE occurrence and so on. I get the idea. By doing that, one would get a continuous date set. However, in the time domain there are gaps due to days without PMSE. Despite of allowing direct comparison with the solar and geomagnetic activities, I would not say that the PMSE data set has become continuous. Still regarding the method, I suggest
adding axis at top showing the time in years in the Figures 5 to 8. This will make easier to follow the time in years.

Despite of positive correlations between PMSE occurrence and solar flux and K index, the authors should point out that the coefficients indicate correlations from moderate to weak.

One important point that the authors have not addressed is the correlation between the duration of the PMSE and the solar and geomagnetic activities.

Minor comments

Page#2, line 2. "Its strongest average echo occurs..." replace by "On average, the strongest echo occurs..."

Page#2, lines 5-6. The sentence "this was recently confirmed by Blix et al. from simultaneous rocket and radar observations (Blix et al., 2003).". I suggest changing it to read as "This was confirmed by Blix et al. (2003) from simultaneous rocket and radar observations."

Page #2, line 8. "...it still provided..." replace by "...it still provides..."

Page #2, lines 14-15. The sentence "...these echoes are a summer phenomenon, lasting from June to August..." may cause some misunderstanding as in the Southern hemisphere is winter. It's better to say clearly which hemisphere those measurements came from.

Page #3, line 7: "...in the same sites..." replace by "...at the same sites..."

Page #3, line 14: "characters" replace by "characteristics"

Page #3, line 25: "...and a cylindrical 120m×46m antenna..." replace by "...and has a cylindrical 120m×46m antenna..."

Page #3, line 26: "...beam-widths of 1.8° north-south and 0.6° east-west was used on

it." I suggest to exclude "was used on it".

Page #4, lines 1-2: I suggest inserting an end point in the sentence "...EISCAT radar." and then start the next one as "The level of electron density...".

Page #5, line 2: To keep the same pattern replace "3-4 kilometers" by "3-4 km"

Page #6, line 22: "we believe" replace by "we consider"

Page #8, line 17: please, inform the order of the polynomial fit.

Page #10, line 7: "lead" replace by "leads"

Page #11, lines 1-2: "...observations shown..." replace by"...observations have shown..."

Page #12, line 8: PMWE replace by PMSE

---

## Editor Comment (EC1) · Igo Paulino (Editor) · 6 Mar 2019

General Comment

The manuscript entitle "Characteristics of layered occurrence ratio of polar mesosphere summer echoes observed by EISCAT VHF 224 MHz radar" by Ge et al. bring an interesting data analysis on the occurrence of strong echoes detected mostly during the summer by radars from polar mesosphere that are called Polar Mesosphere Summer Echoes (PMSE). They have covered a solar cycle of observation from 2004 to 2015 at high latitude (Tromso, Norway).

[Figure]

They have also released important results like (1) non-correlation between the occurrence mono- and double-layers PMSE and solar activity and (2) anti-correlation between the triple-layer PMSE and solar activity.

A priori two reviewers have been nominated to evaluated the paper. Besides, I have some important concerns (listed below) that I would like to be addressed by the authors.

Minor points:

Pg. 1, Throughout the manuscript: earth -> Earth. Pg 2, line 14: …(1) these echoes are summer phenomena. Pg 3, lines 5-6: Please, verify the citations Pg 3, lines 9-11: This statement is confuse, please, re-write it. Pg 4, lines 3-4: The authors have mentioned 6 modes of the radar operation. However, they describe only two of them. Maybe they could explain shortly the difference among all operation modes. Table 3: Please, put the units into the brackets, i.e., (min) instead of /min Pg 7. lines 9-11. Please, give a meaning for the Spearman rank coefficient, in this case. Pg. 16. line 10. "But, we still can not. …"

Major points:

The authors must clarify their contribution with this study. They are using an almost solar cycle of data to study PMSE occurrence and the data is really valuable to understanding some unsolved points on this topic.

Page 8. Line 7. As the author has only o solar cycle, it is not prudent to say that the layered OR has a period of 7-8 year. More data are necessary to conclude about the periodicity that seams to follow the solar activity.

Figures 3 and 4. Why do the authors fit a polynomial curve to the PMSE OR? Is not a sinusoidal curve more appropriated?

Further explanation on Subsection 4.1 and Figures 5, 6 and 7 are necessary. The main point released by the authors was not clear to me, i.e., that there is not direct relation

between the PMSE OR and solar activity.

The same comment above can be extended to Figure 8.

Another concern is regarding to the usage of the threshold to determine the PMSE OR. The authors have not explained why they are using those assumptions. The main conclusion of them are based on these analysis, then it must be clear.

---

## Referee Comment (RC2) · Anonymous Referee #2 · 7 Mar 2019

General comments: Present study characterize the layered polar mesospheric summer echoes (PMSE) occurrence ratio (OR) using EISCAT VHF radar data from 2004-2015. They found that mono and double layer OR is higher than the tri-layer OR, in addition they also noticed a seasonal variation of the OR between these three layers. Further, to estimate the characteristics of the layered PMSE OR a new method has been proposed. Output obtained from the new method is used to understand the solar cycle dependency and geomagnetic variation dependency of the layered PMSE OR. From which it has been concluded that PMSE layered OR is positively correlated with the K index. The correlation between PMSE mono and double- layer OR and F10.7 is relatively weak, and PMSE tri-layer OR has a negative correlation with F10.7.

[Figure]

11 years of EISCAT radar data is used for the characterization, this data covers almost two solar maxima (∼2004 and 2012-2014) and a solar minimum (2006-2009) period. I can understand that the authors put large amount of time and effort to processes the data and the new method. However, I have some serious concern about the new method and the discussion part. In the new method the authors used the altitude information based on this they also try to explain the long term variations (in particular solar and geomagnetic influences on the PMSE OR) but there is no detailed description about this method and how it can take care of the data missing? And the discussion part should be rewritten with citing previous reports and how obtained results are differing from the earlier reports. Thus, I recommend for a major revision prior to the publication.

Specific comments:

1. Section 4.1, the authors introduced a new method for characterize the PMSE OR, they claimed that the new method will avoid the data discontinuity problem? But there is no detailed explanation or justification about how this will compensate the data discontinuity issue? …. Page12, In this section, the day when the first occurrence of PMSE in 2004 (regardless of duration) was recorded as1 and the day with the later occurrence of PMSE increased by sequence…, from these lines what I understood is that they have taken number of occurrence days rather than hours (used in the earlier studies), if it is so, what is the role of altitude and how the OR percentage calculated? Instead of hours if you're taking the number of occurrence by day earlier method (based on time) also may give the same result! Justify it.

2. Figure 2 clearly shows a solar cycle variation, e.g., maximum during solar maxima years and minimum during solar minimum years. But the authors claimed that as a sinusoidal wave! This may mislead the readers. From my understanding if we follow the existing method the influence of solar radiation on PMSE is positive (Bremer et al., 2006). Clarify it.

3. Section 2, There is no a single reference about the EISCAT radar and its data quality! It will be useful if you can include some information about GUISDAP with references. Of course, the radar experiment details are given in table2, however please include the vertical resolution of the data and give brief information about based on which criteria the multiple layers are identified and what is the average occurrence altitude of each layer (i.e., mono, double and tri layer)? 4. To find the characteristic of PMSE occurrence ratio (OR), a computing method and threshold must be defined. First of all, ..., the threshold of electron density (Ne>2.6×1011 m-3) was calculated (Hocking and Röttger ,1997). Not clear, modify the sentence. During the PMSE time the electron density will be bite-out (Kelly 2010) so one can expect decrement in the electron density. Here what the authors meant to say? They have taken only above this limit (Ne>2.6×1011 m-3) or below? 5. It may look good if you change the title as, "Characteristics of layered polar mesospheric summer echoes occurrence ratio observed by EISCAT VHF 224 MHz radar" and discuss about the multiple layered PMSE occurrence and its possible generation mechanism in the introduction part? And brief about why the study of characterization of multiple PMSE OR is important? 6. Page1 line 15, solar cycle, can be used..., modify the sentence. 7. Page1 line 18, PMSE layered..., use only one term either Layer PMSE or PMSE layered throughout the manuscript, my suggestion is use Layered PMSE. 8. Page1 line 20, it can be obtained..., write as, it is obtained..., 9. Page2 line 1, write as, possible indicator of global climate change. 10. Page2 line 5, 2003 is not recent year, change the sentence. 11. Page2 line 7, even though this theory has been presented incompletely..., why? Please give a brief about the incompleteness. 12. Page2 line 23, Yi et al., 2011 citation is irrelevant for this context, they discuss only about the density variation not PMSE. According to Smirnova et al., 2010 F 10.7 is negative but not significant, please mention it. 13. Page3 line 5, spacing are missing 14. Page3 line 11, The correlation of PMSE..., research of 224MHz radar. Sentence not clear. 15. Page3 line 19, The PMSE OR calculation... solve the defects that of measurements...How? What is the demerit of the existing method and how the new method is useful? 16. Antenna beam width in the table and

the text is differs? Write the correct value. 17. Page5 line 6, write as, till now. . ., 18. Section 3.1 modify the subtitle as, Layered PMSE OR calculation method 19. Page6 line 15, . . ., algorithm based on grid partitioning. It will be useful for the readers if you provide little bit detail about this algorithm. 20. In table 3 column 2, is that total observation time for whole year or only the summer time (May-August)? If it is whole year, better to show only from the operation hours of summer months and see is there any difference in the statistics or not? Put the % in row1 and column 6-9, 21. Page8 line 28, write as, explain the occurrence mechanism of PMSE. 22. Page10 line 7, write as, not understood well. 23. Section 4.1, subtitle change as, A new method for layered PMSE OR calculation 24. Page10 line 24, when the PMSE is known to be present. How you decide the PMSE is present or not? Explain it here. 25. Page10 line 24, The ratio between the. . .calculated respectively. Why the ratio is calculated and what is its significance? Brief it. 26. Page12 line 9, We get their variation trends to be largely consistent. . ., rates are reliable. Sentence is not clear. Above the Hocking et al., threshold level the variation is not consistent! Check it. 27. Solar cycle 23, the minimum condition was extended from 2006-2009. 28. Page12 line 15, In other words, no correlation. . ., However, the earlier method shows very clear positive variation with the solar cycle (see figure 2)? Justify it. 29. Page15 line 5, P value less than 0.5, 30. Use the same terminology throughout the manuscript, "either dual layer or double layer, and tri or triple or multi-layer". 31. Page15 line 21, Interestingly, we found that. . ., a negative correlation with F10.7. . ., However, the negative correlation is less than 0.5 and similar kind of result already reported by Smirnova et al. (2010). Why the authors want to highlight this point though the K value also shows similar kind of positive correlation with layer PMSE OR? 32. Page16 line 4, It indicates. . ., how it can indicate? 33. Page16 line 8, the positive correlation between. . ., enhanced magnetic activity caused precipitating particles increase in the mesosphere. Earlier the authors claimed that they removed the precipitation events! 34. Page16 line 8, write as, but still we. . . 35. Page16 line 22, write as, reference or earlier report. 36. Page16 line 23, write as, it is maximum in mid-July. . ., 37. Page16 line 27, under different electron

Interactive
comment
density threshold conditions are largely consistent. I feel above Ne>2.6×1011m-3 this threshold the consistency is not significant (see fig., 9). 38. Page16 line 27, write as, it is found that...,
* * *

---

## Author Comment (AC1) · 28 Mar 2019

The comment was uploaded in the form of a supplement:
https://www.ann-geophys-discuss.net/angeo-2019-13/angeo-2019-13-AC1-supplement.zip

---

## Author Comment (AC2) · 28 Mar 2019

The comment was uploaded in the form of a supplement:
https://www.ann-geophys-discuss.net/angeo-2019-13/angeo-2019-13-AC2-supplement.zip

---

## Author Comment (AC3) · 28 Mar 2019

The comment was uploaded in the form of a supplement: https://www.ann-geophys-discuss.net/angeo-2019-13/angeo-2019-13-AC3-supplement.zip

---

## Author Comment (AC4) · 29 Mar 2019

[revised manuscript text omitted]
 \geq 1 \times 10^{11} m^{-3}$, $N_e \geq 1.5 \times 10^{11} m^{-3}$, $N_e \geq 2.6 \times 10^{11} m^{-3}$, $N_e \geq 3 \times 10^{11} m^{-3}$ and $N_e \geq 3.5 \times 10^{11} m^{-3}$, respectively. We found the variation trends of layered PMSE OR with different threshold are largely consistent. In addition, the larger the threshold, the smaller the ratio. Smirnova et al. (2010) analyzed day-to-day and year-to-year variations of PMSE OR for different thresholds. They found that the choice of the threshold does not influence the shape of the variation curves for PMSE OR. Zeller and Bremer (2009) indicated that different threshold values are for the investigations of the influence of geomagnetic activity on PMSE, however, of less importance. They both think that the variation trends of PMSE OR with different threshold are consistent. The aim of choosing 5 different thresholds is also to increase the number of samples for calculating the correlation coefficients between layered PMSE OR and F10.7 and between layered PMSE OR and K index. Since these occurrence ratios are calculated in the case where the occurrence of PMSE is determined, so, it is recognized that these occurrence rates are reliable. It is well known that the period of 2006-2009 is solar minimum and 2012 is solar maximum, but the PMSE mono- and double-layer average OR in 2007 is not consistent with solar activity. In other words, there is no obvious correlation between mono- and double-layer PMSE OR and solar activity. What's more, we found that PMSE tri- layer OR and solar activity in opposite directions. To prove the conclusion, we will calculate the correlation coefficient between layered PMSE OR and solar activity and between layered PMSE OR and geomagnetic activity in next section. Therefore, the correlation between them can be judged directly.

**45.3 Effect of solar and geomagnetic activity on PMSE OR**

**45.3.1 F$_{10.7}$ index and K -index**

[revised manuscript text omitted]

---

## Author Comment (AC5) · 29 Mar 2019

The comment was uploaded in the form of a supplement:
https://www.ann-geophys-discuss.net/angeo-2019-13/angeo-2019-13-AC5-supplement.pdf

---

## Author Comment (AC6) · 29 Mar 2019

The comment was uploaded in the form of a supplement:
https://www.ann-geophys-discuss.net/angeo-2019-13/angeo-2019-13-AC6-supplement.pdf

---

## Author Response (AR1)

**Dear Editor and referees,**

We are pleased to have been given the opportunity to again revise our manuscript entitled, *"Characteristics of layered occurrence ratio of polar mesosphere summer echoes observed by* EISCAT VHF 224 MHz Radar". We thank you and referees and appreciate the effort of all of you to review our paper and providing us very insightful and constructive comments. Herein we explain how we revised the paper based on reviewer comments and recommendations. We uploaded the following files,

[1] Point-by-Point reply manuscript: in this file replies to comments are given.

[2] Revised Manuscript: this is the clean and 'revised version' of the paper. In this file all the changes made in previously submitted manuscript is 'highlighted' with 'yellow color'.

[3] Track changes manuscript: In this file, there are two kinds of writing:

(a) The 'underline' writing represents the corrected and newly added words and sentences.

(b) The 'strikethrough' writing represents the deleted words and sentences.

We again appreciate the careful review and constructive suggestions of all of you. Below is our reply to comments.

**A point-by-point response to the Editor**

**Reply to Editor's comments:**

**Reply to comment:** before to reply this comment, first the authors would like to thank your careful works and valuable comments. The comments and suggestions are very useful for our manuscript. We have addressed these comments and suggestions, and made (tracked) changes in the manuscript.

**Minor Comments:**

(a): Pg. 1, Throughout the manuscript: earth -> Earth.

reply: It is done. In "Revised Manuscript" we have replaced earth by Earth.

(b): Pg 2, line 14: ... (1) these echoes are summer phenomena.

reply: It is done. In "Revised Manuscript" the correction is at page 2, line 16.

(c): Pg 3, lines 5-6: Please, verify the citations

reply: It is a typo. In "Revised Manuscript" the correction is at line page3, line 7.

(d): Pg 3, lines 9-11: This statement is confused, please, re-write it.

reply: It is done. In "Revised Manuscript" the re-written statement is at page3, line 10-14.

(e): Pg 4, lines 3-4: The authors have mentioned 6 modes of the radar operation. However, they describe only two of them. Maybe they could explain shortly the difference among all operation modes.

**reply:** Thanks for suggestion. We have expanded Table 2 to give the parameters of 6 modes of the EISCAT VHF 224MHz radar.

(f): Table 3: Please, put the units into the brackets, i.e., (min) instead of /min reply: It is done. In "Revised Manuscript" the correction is at Table 3.

(g): Pg 7. lines 9-11. Please, give a meaning for the Spearman rank coefficient, in this case.

reply: It is done. In "Revised Manuscript" the correction is at Pg8, lines 7.

(h): Pg. 16. line10. "But, we still can not. . .."

reply: It is done. In "Revised Manuscript" the correction is at Pg17, lines 27.

**Major Comment:**

(a) The authors must clarify their contribution with this study. They are using an almost solar cycle of data to study PMSE occurrence and the data is really valuable to understanding some unsolved points on this topic.

**reply:** By analyzing the EISCAT VHF radar data, we found that mono and double layer OR is higher than the tri-layer OR. In addition, a seasonal variation of the OR between these three layers is noticed. Furthermore, we have proposed a new method to estimate the characteristics of the layered PMSE OR. Results obtained from this new method is used to understand the solar cycle dependency and geomagnetic variation dependency of the layered PMSE OR. The relationship between layered PMSE OR and  $F_{10.7}$  and between layered PMSE OR and K values also be analyzed. We used the  $F_{10.7}$  and K values corresponding to the occurrence of PMSE with

threshold of  $N_e > 2.6 \times 10^{11} \text{m}^{-3}$ . So that, the correlation of PMSE with solar and geomagnetic

activities is not expected to affect by discontinuous PMSE. The study of relations between PMSE and solar activities and between PMSE and geomagnetic activities are significative.

(b) Page 8. Line 7. As the author has only o solar cycle, it is not prudent to say that the layered OR has a period of 7-8 year. More data are necessary to conclude about the periodicity that seems to follow the solar activity.

**reply:** Thanks for suggestion. Fig.2 shows that the gap between two peaks of PMSE OR is about 7 or 8 years. It is true that we cannot explain that the layered OR has a period of 7-8 year. It is necessary to need more data to conclude about the periodicity that seems to follow the solar activity. We have removed the description form manuscript.

(c) Figures 3 and 4. Why do the authors fit a polynomial curve to the PMSE OR? Is not a sinusoidal curve more appropriated?

**reply:** As described in the paper, Fig. 3 illustrates the mean seasonal variation of the mono-(blue bars) double- (yellow bars) and tri-layer (red bars) PMSE OR and quartic polynomial fitting (black dot-curve) and sine fitting (red dot-dash curve) for the monolayer PMSE OR during 2004-2015. The fitting equation of quartic polynomial fitting is  $f(x) = 1.448 \times 10^{-6} x^4 - 9.715 \times 10^{-4} x^3 + 0.2182 x^2 - 17.82 x + 332.7$  and the fitting degree R=0.5316. The fitting equation of sine fitting is  $f(x) = 23.67 - 11.5 \cdot \cos(0.04509\omega) + 24.79 \cdot sin(0.04509\omega)$  and the fitting degree R=0.5287. According to the fitting results, a quartic polynomial fitting with a relatively high degree of fit is used.

**Fig. 3 Mean seasonal variation of the PMSE mono-(in blue), double-(in yellow), triplelayer (in red) occurrence ratio at Tromsø using observations from 2004 to 2015.**

Fig. 4(a) (b) shows the mean seasonal variation of PMSE total OR (blue bars) and curve-fitting for total PMSE OR during 2004-2015. We used a variety of curve fitting methods. In Fig. 4(a) the fitting equation of gaussian fitting (black dot-curve) is  $f(x) = 86.75 \cdot \exp(-((x-185.2)/32.02)^2)$  and the fitting degree R=0.7579. The fitting equation of cubic polynomial fitting (red dot-dash curve) is  $f(x) = -1.693 \times 10^{-4} x^3 + 0.06584 x^2 - 6.671 x + 125.5$ and the fitting degree R=0.6912. In Fig. 4(b) The fitting equation of  $1/\pi$  harmonic function (green solid curve) is  $f(x) = 41.36-32.72 \cdot \cos(0.05462\omega) \cdot 28.05 \cdot \sin(0.05462\omega)$  and the fitting degree R=0.7714. The fitting equation of  $2/\pi$  harmonic function (pink dash curve) is  $f(x) = 42.37 - 23.39 \cdot \cos(0.0562\omega) - 35.91 \cdot \sin(0.0562\omega) + 5.37 \cdot \cos(0.0562\omega) - 0.3935 \cdot \sin(0.0562\omega)$  and the fitting degree R=0.7816. The fitting equation of  $3/\pi$  harmonic function (yellow dot curve) is  $f(x) = 43.4 - 8.496 \cdot \cos(0.05832\omega) - 42.14 \cdot \sin(0.05832\omega) + 5.826 \cdot \cos(0.05832\omega) + 2.218 \cdot \sin(0.05832\omega)$  and the -5024•cos(0.05832\alpha)-4.666•sin(0.05832\alpha)

fitting degree R=0.7896. According to the fitting degree and the editor's suggestions. We choose the  $3/\pi$  harmonic function fitting. The method is higher goodness of fit and has its applicability.

Fig. 4(a) (b) Mean seasonal variation of the PMSE total occurrence ratio.

(d) Further explanation on Subsection 4.1 and Figures 5, 6 and 7 are necessary. The main point released by the authors was not clear to me, i.e., that there is not direct relation between the PMSE OR and solar activity. The same comment above can be extended to Figure 8. on Subsection 4.1.

**reply:** Thanks for your suggestion. We main study layered PMSE OR in the paper. The legends on the figure4,5,6 is the average of PMSE occurrence rate in three time periods separated by the black dashed line. It is well known that 2006 is solar minimum and 2012 is solar maximum, the legends on the figure4,5,6 shown the PMSE mono- and double-layer average OR is not consistent with solar activity. So, we say that there has no correlation between PMSE monoand double-layer OR and solar activity. To prove the conclusion, we calculate the correlation coefficient between PMSE layered OR and solar activity and between PMSE layered OR and geomagnetic activity in next section. Then the conclusion is convinced. We have made improvements to make Fig5-8 easier to understand in revised manuscript.

(e) Another concern is regarding to the usage of the threshold to determine the PMSE OR. The authors have not explained why they are using those assumptions. The main conclusion of them

**are based on these analysis, then it must be clear.**

**reply:** Thanks for your suggestion. In order to obtain the correlation between mono, double and triple layer PMSE OR, we defined 5 electron density thresholds. Of course, you can define other threshold values. Smirnova et al. (2010) found that the choice of the threshold does not influence the shape of the variation curves for PMSE OR. Zeller and Bremer (2009) indicated that different threshold values are for the investigations of the influence of geomagnetic activity on PMSE, however, of less importance. Because, we will calculate the correlation coefficients between layered PMSE OR and  $F_{10.7}$  and between layered PMSE OR and K index. The aim of choosing 5 different thresholds is to increase the number of samples for correlation coefficient calculations. We give a more detailed explanation in revised manuscript at page13, line 6-16 for this problem.

**Reference**

- Smirnova, M., Belova, E., Kirkwood, S., and Mitchell, N.: Polar mesosphere summer echoes with ESRAD, Kiruna, Sweden: Variations and trends over 1997–2008, Journal of Atmospheric and Solar-Terrestrial Physics, 72, 435-447, doi:10.1016/j.jastp.2009.12.014, 2010.
- Zeller O. and Bremer J., The influence of geomagnetic activity on mesospheric summer echoes in middle and polar latitudes, Annales Geophysicae, 27(2): 831-8372, DOI: 10.5194/angeo-27-831-2009, 2009.

**A point-by-point response to the Referee#1**

**Reply to Reviewer#1's comments:**

**Reply to comment:** before to reply this comment, first the authors would like to thank the reviewer for guidance. The reply to this comment is given stepwise here, because we want to show the mistake and also its correction.

**Specific comments:**

(a): At page #1, lines 1-2: the sentence: "The ionosphere is an important part of the near the earth space environment and the mesosphere is the coldest region in the earth's atmosphere at local summer time." Regardless the season the mesosphere is the coldest region of the Earth's atmosphere, not only during the summer. I suggest rewrite the sentence to make this clearer.

**reply:** Thanks to your suggestion. We have revised this as "The ionosphere is an important part of near the Earth space environment and the mesosphere is the coldest region in the Earth's atmosphere". In revised manuscript it can be found at page#1, lines29.

(b): Section 3.1-Calculation method: The authors should explain better the reason to use the threshold of  $2.6 \times 10^{11}$  electrons/m3 to detect the PMSE.

**reply:** Thanks to your suggestion. We added a better explanation in the revised manuscript. Volume reflectivity is defined as "backscattering cross section per unit volume" (Hocking,

1985). Noted that:  $\eta = \sigma_0 \times N_e$ , where  $\eta$  is the volume reflectivity,  $\sigma_0 = 5 \times 10^{-29} m^2$

is the effective scattering cross section, and  $N_e$  is the electron density (raw electron density can represent equivalent electron density for the case of PMSE) measured by the EISCAT radars. The selection of PMSE threshold is still an open question. Different threshold has been used for detecting PMSE echoes by VHF radar. For example, see the Table 1 given below. We used the PMSE threshold given by Hocking and Röttger (1997). The reason for using  $N_e=2.6\times10^{11}$  m-3 as threshold is that it corresponds to the threshold ( $\eta=1.3\times10^{-17}$  m-1) used for PMSE. Therefore, in this study the PMSE were considered to be present only if the electron density satisfies the threshold ( $N_e > 2.6\times10^{11}$  m-3).

| Frequency (Bragg scale)
MHz (m) | Location       | Reference                  | Reflectivity                                 |
|------------------------------------|----------------|----------------------------|----------------------------------------------|
|                                    |                | Hoppe et al. (1988)        | $1.5 \times 10^{-16}$                        |
|                                    |                | Röttger and LaHoz (1990)   | 2.3×10 -17                        |
| 224(0.67)                          | Tromsø (69° N) | Hocking and Röttger (1997) | 1.3×10 -17 -1.3×10 -15 |
|                                    |                | Belova et al. (2007)       | $1.5 \times 10^{-14}$                        |
|                                    |                | Rapp et al. (2008)         | $5.0 \times 10^{-14}$                        |

| Table1: | PMSE    | studied  | with | calibrated | radars | at | 224MHz. | This | table | is | referenced | from | Li |
|---------|---------|----------|------|------------|--------|----|---------|------|-------|----|------------|------|----|
| (2011). | (see Ap | pendix A | A)   |            |        |    |         |      |       |    |            |      |    |

(c): At page #6, lines 11-13: In that sentence the authors mention a condition  $t \ge 1$  min. It is not clear where this condition came from. They should make this clearer.

**reply:** Thanks to your suggestion. For calculating the PMSE OR, we have selected only those events for which the PMSE threshold ( $N_e > 2.6 \times 10^{11} \ m^{-3}$ ) is satisfied, for time (t  $\geq 1 \ min$ )

in the altitude range of 80 – 90 km. Of course, you can also define the time  $t \ge$  any time interval.

(d): The description of the method of calculation at page #6, lines 15-20, which takes as an example of the monolayer PMSE occurrence, seems to be a little confused. The description is clearer when the authors described the occurrence ratio of the double and tri-layer PMSE. I suggest to rewrite the description of the monolayer PMSE occurrence. From Table 3 one can see that the author defined the OR of the PMSE as the percentage ratio between the duration of the mono, double and triple layer PMSE and the total time of observations. The description mentioned above should be as clear as the information coming from the Table 3.

**reply:** Thanks for suggestion. We have revised the abovementioned description as "The calculation method is based on individual horizontal profiles. When the electron density satisfy the PMSE threshold  $N_e > 2.6 \times 10^{11}$  m-3, then that time was taken as the starting time of the PMSE occurrence and the time when the electron density fails to satisfy the threshold was taken as the end time of PMSE occurrence. The time of PMSE duration is the time difference between the end and the starting time of the PMSE occurrence. Taking the calculation method of PMSE monolayer occurrence ratio as an example: We defined the ratio between the sustained time of monolayer PMSE and the total observation time as the PMSE monolayer OR." The applied procedure for the detection of multiple PMSE layers is based on individual vertical profiles with a high temporal resolution (Hoffmann, P. 2004). The layer ranges are identified by an electron density threshold of  $2.6 \times 10^{11}$ m-3 ( $N_e > 2.6 \times 10^{11}$ m-3). Once a vertical profile of the electron density has two peaks and these two peaks are higher than the threshold ( $N_e > 2.6 \times 10^{11}$ m-3), we select it as a double layer. The PMSE double-layer OR is the ratio between the sustained time of PMSE double-layer OR is also calculated in this

**way.". In revised manuscript this can be found at page#6, lines 14-15 and page#7, line1.**

(e): In section 4, the authors propose a method to make PMSE OR continuous. They considered as day 1 the first PMSE occurrence in 2004, day 2 as second PMSE occurrence and so on. I get the idea. By doing that, one would get a continuous date set. However, in the time domain there are gaps due to days without PMSE. Despite of allowing direct comparison with the solar and geomagnetic activities, I would not say that the PMSE data set has become continuous. Still regarding the method, I suggest adding axis at top showing the time in years in the Figures 5 to 8. This will make easier to follow the time in years.

**reply:** Thanks for suggestion. We used  $F_{10.7}$  values and geomagnetic K index values corresponding to the occurrence of PMSE. That is, when PMSE events occurred on the day, we took the  $F_{10.7}$  and K index values for this day. If there is no PMSE, we will not take the values of  $F_{10.7}$  and K index. Because we analyze the variations of PMSE mono-, double- and triple-layer OR with threshold conditions of  $N_e > 1 \times 10^{11} \text{m}^{-3}$ ,  $N_e > 1.5 \times 10^{11} \text{m}^{-3}$ ,  $N_e > 2.6 \times 10^{11} \text{m}^{-3}$ ,  $N_e > 3 \times 10^{11} \text{m}^{-3}$  and  $N_e > 3.5 \times 10^{11} \text{m}^{-3}$  during 2004-2015, the number of PMSE events in the same year is different with different threshold conditions. It is possible to happen: such as in 2004, the PMSE case occurred 10 times under the threshold conditions of  $N_e > 3.5 \times 10^{11} \text{m}^{-3}$ , and the PMSE case occurred 8 times under threshold conditions of  $N_e > 3.5 \times 10^{11} \text{m}^{-3}$ . Therefore, we can't add axis at top showing the time in years in the existing Figures 5 to 7. However, we redrew Figures 5 to 7 and adding axis at top showing the time in years. In this way, the relationship between Figure 5-6 and Figure 8 becomes clear. In revised manuscript this can be found at

**page#12 and 15, Figs.5,6,7,8.**

(f): Despite of positive correlations between PMSE occurrence and solar flux and K index, the authors should point out that the coefficients indicate correlations from moderate to weak. reply: Thanks for suggestion. According to the Referee's advice, we have revised them and it

can be found at page#17, lines 6-7 and 10.

(g): One important point that the authors have not addressed is the correlation between the duration of the PMSE and the solar and geomagnetic activities.

**reply:** Thank you for valuable comments. Because PMSE echoes are intermittent. The duration of PMSE is very short, some are only a few minutes.  $F_{10.7}$  value is the average data of the day, the K index value is the average data of 3 hours, so the correlation between the duration of the PMSE and the solar and geomagnetic activities are still not discussed. But we will continue to do our best to solve this problem.

**Minor Comments:**

(a): Page#2, line 2. "Its strongest average echo occurs..." replace by "On average, the strongest echo occurs..."

reply: It is done. In "Revised Manuscript" the correction is at Page#2, line 3.

(b): Page#2, lines 5-6. The sentence "this was recently confirmed by Blix et al. from simultaneous rocket and radar observations (Blix et al., 2003).". I suggest changing it to read as "This was confirmed by Blix et al. (2003) from simultaneous rocket and radar observations." **reply:** It is done. In "Revised Manuscript" the correction is at Page#2, lines 5-6.

(c): Page #2, line 8. "...it still provided..." replace by "...it still provides..."

**reply:** In "Revised Manuscript" the description was removed after think with care.

(d): Page #2, lines 14-15. The sentence "...these echoes are a summer phenomenon, lasting from June to August..." may cause some misunderstanding as in the Southern hemisphere is winter. It's better to say clearly which hemisphere those measurements came from.

reply: It is done. In "Revised Manuscript" the correction is at Page#2, lines 16-17.

(e): Page #3, line 7: ". . . in the same sites. . ." replace by ". . . at the same sites. . ."

reply: It is done. In "Revised Manuscript" the correction is at Page#3, lines 9.

(f): Page #3, line 14: "characters" replace by "characteristics"

reply: It is done. In "Revised Manuscript" the correction is at Page#3, line 16.

(g): Page #3, line 25: ". . .and a cylindrical 120m×46m antenna. . ." replace by ". . .and has a cylindrical 120m×46m antenna.

reply: It is done. In "Revised Manuscript" the correction is at Page#3, line 28.

(h): Page #3, line 26: ". . .beam-widths of 1.8° north-south and 0.6° east-west was used on it." I suggest to exclude "was used on it".

reply: In "Revised Manuscript" the description was rewrote after think with care.

(i): Page #4, lines 1-2: I suggest inserting an end point in the sentence ". . .EISCAT radar." and then start the next one as "The level of electron density. . .".

reply: It is done. In "Revised Manuscript" the correction is at Page#5, line 5.

(i): Page #5, line 2: To keep the same pattern replace "3-4 kilometers" by "3-4 km"

reply: It is done. In "Revised Manuscript" the correction is at Page#5, line 17.

(k): Page #6, line 22: "we believe" replace by "we consider"

reply: It is done. In "Revised Manuscript" we have rewrote it, at page#7, line 1-6.

(1): Page #8, line 17: please, inform the order of the polynomial fit.
reply: It is done. In "Revised Manuscript" the correction is at Page#9, line1.
(m): Page #10, line 7: "lead" replace by "leads"
reply: It is done. In "Revised Manuscript" the correction is at Page#10, line 15.
(n): Page #11, lines 1-2: "...observations shown..." replace by "...observations have shown..."
reply: It is done. In "Revised Manuscript" the correction is at Page#11, line 15.
(o): Page #12, line 8: PMWE replace by PMSE
reply: It is done. In "Revised Manuscript" the correction is at Page#16, line 10.

**References:**

- Belova, E., P. Dalin, and S. Kirkwood, Polar mesosphere summer echoes: A comparison of simultaneous observations at three wavelengths, Ann. Geophys., 25, 2487–2496, doi: org/10.5194/angeo-25-2487-2007, 2007.
- Hocking, W. K., Measurement of turbulent energy dissipation rates in the middle atmosphere by radar techniques: A review, Radio Sci., 20, 1403–1422, doi:10.1029/RS020i006p01403, 1985.
- Hocking, W. K., and J. Röttger, Studies of polar mesosphere summer echoes over EISCAT using calibrated signal strengths and statistical parameters, Radio Sci., 32, 1425–1444, doi:10.1029/97RS00716, 1997.
- Qiang, L., Multi-frequency radar observations of polar mesosphere summer echoes: Statistical properties and microphysical results, INAUGURAL- DISSERTATION, 2011.
- Hoppe, U.P., C. Hall, and J. R"ottger, First observations of summer polar mesospheric backscatter with a 224 MHz radar, Geophys. Res. Lett., 15, 28–31, doi:10.1029/GL015i001p00028, 1988.
- Rapp, M., I. Strelnikova, R. Latteck, P. Hoffman, U.-P. Hoppe, I. Häggström, and M. Rietveld, Polar mesosphere summer echoes (PMSE) studied at Bragg wavelengths of 2.8 m, 67 cm, and 16 cm, J. Atmos. Sol. Terr. Phys., doi: 10.1016/j.jastp.2007.11.005, 2008.
- Röttger, J., and C. LaHoz, Characteristics of polar mesosphere summer echoes (PMSE) observed with the EISCAT 224 MHz radar and possible explanations of their origin, J. Atmos. Terr. Phys., 52, 893–906, doi:10.1016/0021-9169(90)90023-G, 1990.

**A point-by-point response to the Referee#2**

**Reply to Referee#2's comments:**

**Reply to comment:** before to reply this comment, first the authors would like to thank your careful works and valuable comments. The comments and suggestions are very useful for our manuscript. We have addressed these comments and suggestions, and made (tracked) changes in the manuscript.

**Specific Comments:**

(1): Section 4.1, the authors introduced a new method for characterize the PMSE OR, they claimed that the new method will avoid the data discontinuity problem? But there is no detailed explanation or justification about how this will compensate the data discontinuity issue? .... Page12, In this section, the day when the first occurrence of PMSE in 2004 (regardless of duration) was recorded as1 and the day with the later occurrence of PMSE increased by sequence. . ., from these lines what I understood is that they have taken number of occurrence days rather than hours (used in the earlier studies), if it is so, what is the role of altitude and how the OR percentage calculated? Instead of hours if you're taking the number of occurrences by day earlier method (based on time) also may give the same result! Justify it.

**reply:** The day when the first occurrence of PMSE in 2004 (regardless of duration) was recorded as 1, and the day with the later occurrence of PMSE increased by sequence. A contiguous array was obtained, then take  $F_{10.7}$  and the median of the K index during a day values corresponding to the occurrence of PMSE, which is also a continuous array. Next, we discuss the correlation between layered PMSE OR and  $F_{10.7}$  and between layered PMSE OR and K values. Since the occurrence of PMSE is not continuous during the day, sometimes the occurrence is very short (a few minutes). It is very difficult to discuss the relationship between PMSE OR, solar and geomagnetic activity Without this method. We used the  $F_{10.7}$  and geomagnetic K index where PMSE occurrence, there is a corresponding relationship between PMSE and  $F_{10.7}$  and between PMSE and K index. If so, they are correlativity. In the long term, their relationship is convincing.

The second method for calculating PMSE OR: First of all, a computing threshold of electron density is defined. We have specified a certain altitude range and the observation time of the radar is known, which constitutes a rectangular area. Calculate the number of electron density  $N_e > 2.6 \times 10^{11}$  m-3 and the total number of electron density in this area, the ratio of them is PMSE OR. That is, PMSE OR=the number of electron density  $N_e > 2.6 \times 10^{11}$  m-3 / the number of total electron density.

The first method for calculating PMSE OR: The applied procedure is based on individual horizontal profiles. When  $N_e > 2.6 \times 10^{11} \text{ m}^{-3}$ , the time is taken as the starting time of the PMSE occurrence time; When  $N_e \le 2.6 \times 10^{11} \text{ m}^{-3}$  with horizontal stacking time sections, the time is the end time of PMSE. Layered PMSE OR= the sustained time of layered PMSE / the total observation time of radar. PMSE OR is different by the two calculation methods and the multi-layer PMSE OR calculated by the second method is higher than the first method. But there is no right or wrong between the two methods, the definition of calculation method is different. Identified on multi-layer PMSE: There is alternations between electron density  $> 2.6 \times 10^{11} \text{ m}^{-3}$

and  $< 2.6 \times 10^{11}$ m-3 at vertical altitude. We identify that there are multiple layered PMSE. The specific distinguish of double layer or triple layers of PMSE, it depends on the number of PMSE layer were increased with the increase of times of the electron density  $> 2.6 \times 10^{11}$ m-3 replace the electron density  $< 2.6 \times 10^{11}$ m-3 at the exact same time. We first determine whether the echo is a mono-layer PMSE or double-layer PMSE and then calculate the PMSE OR.

(2): Figure 2 clearly shows a solar cycle variation, e.g., maximum during solar maxima years and minimum during solar minimum years. But the authors claimed that as a sinusoidal wave! This may mislead the readers. From my understanding if we follow the existing method the influence of solar radiation on PMSE is positive (Bremer et al., 2006). Clarify it.

**reply:** Thanks for suggestion. It may be some misunderstood. The sinusoidal wave that we are talking about is not the relationship between the solar activity and layered PMSE, but the trend of mono- double- and triple-layer PMSE OR, which has obvious wave peak and wave valley. If it can be confirmed that layered PMSE OR is closely linearly related to solar activity, then the trends of PMSE OR should be periodical, so we did the following correlation analysis. Smirnova et al. (2010) shows the correlation of the year-by-year variations of PMSE occurrence rate and length of season with solar activity, represented by the solar 10.7 cm radio flux, is negative but not significant. This is consistent with our results, but contrary to the result of Bremer et al., (2006). Therefore, it is still a scientific project worth exploring.

(3): Section 2, There is no a single reference about the EISCAT radar and its data quality! It will be useful if you can include some information about GUISDAP with references. Of course, the radar experiment details are given in table2, however please include the vertical resolution of the data and give brief information about based on which criteria the multiple layers are identified and what is the average occurrence altitude of each layer (i.e., mono, double and tri layer)?

**reply:** The EISCAT VHF (224 MHz) radars are collocated at Tromsø, Norway (69.61N, 19.21E). It is powerful tool for studying the lower ionosphere. Detailed descriptions of the radar can be found in Baron (1986). These measurements by EISCAT radar are very well suited for investigating the characteristics of PMSE. (for previous work, see e.g. Li et al., 2010 and references therein). In our case, the analysis was done using the well documented 'GUISDAP' software package and taking into account measurements with the local ionosonde (see Lehtinen and Huuskonen, 1996 and www.eiscat.se for details) The data acquisition channels of the radar start at 59.7 km and up to 139.5 km with a range resolution of 300 m (i.e., height resolution owing to the radar beam vertically pointing for all the observations) the altitude resolution is include in table1.

Identified on multi-layer PMSE: The applied procedure for the detection of multiple PMSE layers is based on individual vertical profiles with a high temporal resolution. The layer ranges are identified by an electron density threshold of  $2.6 \times 10^{11} \text{m}^{-3}$  ( $N_e > 2.6 \times 10^{11} \text{m}^{-3}$ ). Once a vertical profile of the electron density has two peaks and these two peaks are higher than the threshold ( $N_e > 2.6 \times 10^{11} \text{m}^{-3}$ ), we select it as a double layer. For a detailed instruction on multiple structures see e.g. (Hoffmann, P. 2005 and Ge et al. 2016).

---

## Referee Report (RR1)

Comments on "**Characteristics of layered polar mesosphere summer echoes occurrence ratio observed by EISCAT VHF 224MHz Radar**" by Ge et al.,

The revised manuscript has improved a lot from the previous version of the manuscript. I acknowledge the authors hard work. However, still I am not convinced with some of the responses. For example, in figure2 the multiple PMSE occurrence show maximum during solar maximum and minimum during solar minimum (positive variation kind of structure) but the correlation is negative why? After the clarification this manuscript may accept for the publication in AnGeo Physicae. So, I recommend for a minor revision.

Comments:

Page 9 line 18-21, Table 3 indicates a difference in total observation time for the individual years. How has this been taken into account for the determination of occurrence ratios? …., whatever the time is different, it is clear (from the percentage) that the multiple PMSE-OR rate is less during the solar minimum years (2006-09)! Comment on it?

Reply to Question no:2. If it can be confirmed that layered PMSE OR is closely linearly related to solar activity, then the trends of PMSE OR should be periodical, so we did the following correlation analysis. Smirnova et al. (2010) shows the correlation of the year-by-year variations of PMSE occurrence rate and length of season with solar activity, represented by the solar 10.7 cm radio flux, is negative but not significant. This is consistent with our results!

But in the manuscript the correlation coefficient as follows: page 8 line: 7-8, **the correlation coefficients ($rs$) of mono-layer with double-layer OR, tri-layer OR and total OR are 0.7922, 0.7718 and 1, respectively.** The correlation coefficient is very high! It simply means that it have positive correlation with solar cycle variation. However, the authors claimed that it is a negative correlation, why? Please either modify the text or give the appropriate evidence that the multiple PMSE OR have negative correlation with solar cycle variation.

Page 5 line 21, PMSE occur in thin layers having thickness up to 3-4 km. what is the average thickness of the single, double and triple layer? I feel during the multiple PMSE occurrence time the thickness will decrease. Is it so?

Reply 26, please check the greater than symbol, I think it would be less than (<) is the second range. For example, Ne> $1x10^{-11}$ m$^{-3}$ and less than Ne<$1.5x10^{-11}$ m$^{-3}$. Please check it.

---

## Author Response (AR2)

**Dear Editor,**

We are pleased to have been given the opportunity to again revise our manuscript entitled, *"Characteristics of layered occurrence ratio of polar mesosphere summer echoes observed by EISCAT VHF 224 MHz Radar".* We appreciate the effort of all of you to review our manuscript

5   and providing us very insightful and constructive comments. Herein we explain how we revised the manuscript based on reviewer comments and recommendations.

We uploaded the following files,

[1] Point-by-Point reply manuscript: in this file replies to comments are given.

[2] Revised Manuscript: this is the clean and 'revised version' of the paper.

10  [3] Track changes manuscript: In this file, there are two kinds of writing:

(a) The 'underline' writing represents the corrected and newly added words and sentences.

(b) The 'strikethrough' writing represents the deleted words and sentences.

**Dear Reviews,**

15  Before to reply to this comment, first the authors would like to thanks your careful works and valuable comments. The comments and suggestions are very useful for our manuscript. We have addressed these comments and suggestions and made (tracked) changes in the manuscript.

**Reply to comments of Referee #1:**

20  Thanks for your suggestions. We have revised grammar issues in the revised manuscript.

**Reply to comments of Referee #2:**

**Comments:**

**(1):** Page 9 line 18-21, Table 3 indicates a difference in total observation time for the individual

25  years. How has this been taken into account for the determination of occurrence ratios? ...., whatever the time is different, it is clear (from the percentage) that the multiple PMSE-OR rate is less during the solar minimum years (2006-09)! Comment on it?

**reply:** We downloaded the PMSE data from the website. (https://www.eiscat.se/schedule/schedule.cgi?year=2004&month=7&S=on&A=on&VHF=on

30  &HEA=on). The total observation time for individual year is different because the EISCAT VHF radar observation is discontinuous. Most of the previous papers are the results of analyzing continuous data of the MST radar, which is different from the data types in this manuscript. In order to reduce the impact of discontinuous data, we present a new method which is to extract discontinuous data separately, then we calculate the PMSE OR. We find the corresponding

35  background parameters and analyze the relationship between them in this manuscript. All the work is based on the characteristics of the EISCAT radar data, then we find a better method and get a more credible conclusion. Table 3 shows the annual mean of PMSE OR. Since, we have calculated the occurrence ratio in different years individually, so the difference in the total observations time does not affect the occurrence ratio.

40  From the situation that the annual mean multiple PMSE-OR is less during the solar minimum years (2006-09) and maximum during solar maximum, it is clear that annual mean layered PMSE-OR is positively correlated with $F_{10.7}$. But we think it is not inconsistent with the

negative correlation obtained on page 17 line:9-10. On page 17 line: 9-10, we found that the PMSE tri-layer OR has a negative correlation with $F_{10.7}$. Herein, The PMSE OR is the OR of every discontinuous PMSE case unlike the annual mean PMSE OR shown in Fig. 2. We sort out each PMSE case and arrange them in chronological order. It can get a data series of PMSE OR that are completely different from those in Fig. 2.

**(2):** Reply to Question no:2. "If it can be confirmed that layered PMSE OR is closely linearly related to solar activity, then the trends of PMSE OR should be periodical, so we did the following correlation analysis. Smirnova et al. (2010) shows the correlation of the year-by-year variations of PMSE occurrence rate and length of season with solar activity, represented by the solar 10.7 cm radio flux, is negative but not significant. This is consistent with our results." But in the manuscript the correlation coefficient as follows: page 8 line: 7-8, the correlation coefficients ($r_s$) of mono-layer with double-layer OR, tri-layer OR and total OR are 0.7922, 0.7718 and 1, respectively. The correlation coefficient is very high! It simply means that it has positive correlation with solar cycle variation. However, the authors claimed that it is a negative correlation, why? Please either modify the text or give the appropriate evidence that the multiple PMSE OR have negative correlation with solar cycle variation.

**reply:** First of all, we regret we made one mistake about the correlation coefficients. On page 8 line: 8-9, we have recalculated the correlation coefficients of mono-layer with double-layer OR, tri-layer OR and total OR, having values of 0.7922, 0.7718 and 0.9480, respectively. Since the data measures by EISCAT VHF radar are discontinuous, and PMSE only occurs for a few days each year. Herein, The OR is the annual mean OR. It simply means that annual mean PMSE mono-layer OR is positively correlated with annual mean double-layer OR, annual mean tri-layer OR and annual mean total OR. This correlation has nothing to do with solar cycle variation.

The claim about the negative correlation is on page 17 line: 9-10. We found that the PMSE tri-layer OR has a negative correlation with $F_{10.7}$. Herein, The PMSE OR is the OR of every discontinuous PMSE case unlike the annual mean PMSE OR shown in Fig. 2. We sort out each PMSE case and arrange them in chronological order. It can get a data series of PMSE OR that are completely different from those in Fig. 2. We analyze the correlation of multiple PMSE OR with $F_{10.7}$ which corresponding to every occurrence of PMSE. Therefore, even if there is a positive correlation on page 8, it is reasonable to be a negative correlation on page 17.

**(3):** Page 5 line 21, PMSE occur in thin layers having thickness up to 3-4 km. what is the average thickness of the single, double and triple layer? I feel during the multiple PMSE occurrence time the thickness will decrease. Is it so?

**reply:** The 3-4km described herein is the average thickness of the monolayer. We have already revised it in the manuscript. In fact, the average thickness of double-layer is the same as mono-layer, but the average thickness of tri-layer is different. For example, Fig. 1(a) shows the typical events of PMSE double-layer, and we can find the average thickness of its every layer is 3-4km. Fig. 1(b) shows the typical events of PMSE tri-layer, and we can see that the average thickness of every layer is decreasing.

[Figure]

(a) (b)

Fig.1 (a) Double layer PMSE. (b) Tri-layer PMSE.

**(4)** please check the greater than symbol, I think it would be less than (<) is the second range. For example, Ne> $1 \times 10^{-11}$ m$^{-3}$ and less than Ne<$1.5 \times 10^{-11}$ m$^{-3}$. Please check it.

**reply:** Thanks for the suggestion. For example, in Fig.9 we calculated the Pearson linear correlation coefficients between monolayer PMSE OR with threshold $N_e > 1 \times 10^{11}$m$^{-3}$ and $F_{10.7}$, and between monolayer PMSE OR with threshold $N_e > 1.5 \times 10^{11}$m$^{-3}$ and $F_{10.7}$. Herein, it's not necessary for $N_e$ to be less than (<) the second range($1.5 \times 10^{11}$m$^{-3}$).

[revised manuscript text omitted]